# Insights into the Cytochrome P450 Monooxygenase Superfamily in *Osmanthus fragrans* and the Role of *OfCYP142* in Linalool Synthesis

**DOI:** 10.3390/ijms232012150

**Published:** 2022-10-12

**Authors:** Jiawei Liu, Hongmin Hu, Huimin Shen, Qingyin Tian, Wenjie Ding, Xiulian Yang, Lianggui Wang, Yuanzheng Yue

**Affiliations:** 1Key Laboratory of Landscape Architecture, Jiangsu Province, College of Landscape Architecture, Nanjing Forestry University, Nanjing 210037, China; 2Co-Innovation Center for Sustainable Forestry in Southern China, Nanjing Forestry University, Nanjing 210037, China

**Keywords:** cytochrome P450 monooxygenase, fragrance, linalool, *Osmanthus fragrans*

## Abstract

*Osmanthus fragrans* flowers have long been used as raw materials in food, tea, beverage, and perfume industries due to their attractive and strong fragrance. The P450 superfamily proteins have been reported to widely participate in the synthesis of plant floral volatile organic compounds (VOCs). To investigate the potential functions of P450 superfamily proteins in the fragrance synthesis of *O*. *fragrans*, we investigated the P450 superfamily genome wide. A total of 276 *P450* genes were identified belonging to 40 families. The RNA-seq data suggested that many *OfCYP* genes were preferentially expressed in the flower or other organs, and some were also induced by multiple abiotic stresses. The expression patterns of seven flower-preferentially expressed *OfCYP*s during the five different flower aroma content stages were further explored using quantitative real-time PCR, showing that the CYP94C subfamily member *OfCYP142* had the highest positive correlation with linalool synthesis gene *OfTPS2*. The transient expression of *OfCYP142* in *O*. *fragrans* petals suggested that *OfCYP142* can increase the content of linalool, an important VOC of the *O*. *fragrans* floral aroma, and a similar result was also obtained in flowers of *OfCYP142* transgenic tobacco. Combined with RNA-seq data of the transiently transformed *O*. *fragrans* petals, we found that the biosynthesis pathway of secondary metabolites was significantly enriched, and many 2-C-methyl-D-erythritol 4-phosphate (MEP) pathway genes were also upregulated. This evidence indicated that the OfCYP proteins may play critical roles in the flower development and abiotic response of *O*. *fragrans*, and that *OfCYP142* can participate in linalool synthesis. This study provides valuable information about the functions of *P450* genes and a valuable guide for studying further functions of *OfCYP*s in promoting fragrance biosynthesis of ornamental plants.

## 1. Introduction

The cytochrome P450 (CYP) monooxygenases, known as multifunctional oxidases, are encoded by a complex gene superfamily. This was the first group of enzymes classified as a “super family” and named after CYP because of the absorption peak at 450 nm produced by the combination of P450 and reductive carbon monoxide [1,2]. A large number of *CYP* genes have been reported in many plants including wheat (*Triticum aestivum*), castor (*Ricinus communis*), and avocado (*Persea americana*) fruit [3,4,5,6,7]. Based on the similarities of the amino acid sequences encoded by the genes and the criterion of phylogeny and homology, the *CYP* genes are divided into 11 clans: seven single-family clans (CYP51/74/97/710/711/727/746) and four multiple-family clans (CYP71/72/85/86). Each clan is named according to the lowest family number in the group [2,8,9]. Each CYP protein has about 500 amino acids, and the tertiary construction of the protein is generally configured by 13 alpha helices and four beta sheets. At present, determining if a gene belongs to the CYP450 gene family mainly depends on whether there is a conserved heme-binding domain (FxxGxRxCxG) in its structure. If this exists, the gene can be grouped into the CYP450 gene family [10]. Over 20,000 CYP gene family members have been characterized but the functions of most remain to be revealed [8].

The CYP proteins with a broad range of catalytic activity are distributed in all branches in the evolution of life and participate in various secondary metabolic processes of plants [2]. Plant terpenoids have an impressive structural diversity. Among them, linalool, ocimene, *cis*-linalool oxide, and *trans*-linalool oxide make major contributions of volatiles to the fragrance of sweet osmanthus (*Osmanthus fragrans*) [11,12]. Complex combinations of terpenes are the common constituents of the characteristic aromas of different cultivars [13]. Various volatile terpenoids are formed directly by catalysis using terpenoid synthases (TPSs). Previous studies provided evidence that cytochrome P450 enzymes involved in monoterpenoid metabolism are usually co-expressed with TPSs [14,15]. Gene co-expression analysis was used to predict which *P450* genes were involved in monoterpenoid metabolism [16]. The expression of *CYP76C1* from Arabidopsis (*Arabidopsis thaliana*) is tightly coregulated with that of *TPS10* and *TPS14*. These genes work together to promote and modulate the emission of linalool and the production of soluble carboxylic linalool in opening flowers [17]. *AtTPS10* and *AtTPS14* were also co-expressed in the same plant tissue with *AtCYP71B31* and *AtCYP76C3*, respectively, and this tight co-expression pattern reveals linalool-coupled production and oxidation in Arabidopsis flowers [16]. Remarkably, *AtCYP71B31* and *AtCYP76C3*, which belong to distinct subfamilies, are important regulators in oxidative metabolism of both (R)- and (S)-enantiomers of linalool in Arabidopsis [16]. In addition, other members of the CYP76C subfamily from Arabidopsis such as *CYP76C2* and *CYP76C4* are also involved in linalool metabolism, synthesizing the main product 8-hydroxylinalool with 9-hydroxylinalool formed as a minor product [18].

Linalool and linalool derivatives not only comprise floral fragrance but also perform defense against floral antagonists [19,20]. Many *CYP* genes also participate in stress responses, including to biological and abiotic stress. In rice, *CYP94C2b* promotes jasmonic acid metabolism and endows plant salinity tolerance [21]. In Arabidopsis, *AtCYP709B1*, *AtCYP709B2*, and *AtCYP709B3* play roles in salt and abscisic acid stress [22]; and *AtCYP79B2*, *AtCYP79B3*, *AtCYP71A12*, *AtCYP71A13*, and *AtCYP71B15* are involved in synthesis of camalexin–a phytoalexin with antibacterial, antifungal, and antiproliferative activities [23,24,25,26]. Many studies have focused on the *CYP* genes that play critical roles in plant stress responses and the synthesis of floral volatile organic compounds (VOCs) [16,21,27]. However, no monoterpene-synthesizing cytochrome P450 in *O*. *fragrans* has been reported thus far.

Sweet osmanthus has excellent fragrance and is one of the most famous evergreen plants [28,29]. Its unique sweet fragrance is one of its most important ornamental traits. Some VOCs responsible for the aroma, especially linalool, have important economic and commercial value and are widely used as fragrances in foods and beverages [30,31]. The genomic data of sweet osmanthus provides the opportunity for systematic research to characterize the *P450* gene family members [32]. To further characterize the *CYP* genes with putative function, we compared the *CYP* genes in *O. fragrans* with those in Arabidopsis by analysis of their phylogenetic relationship. Analyses of structure, conserved motifs, and expression patterns of *OfCYP* genes were also conducted. Among the seven preferentially expressed *OfCYP*s in flowers, *OfCYP142* was screened as a candidate gene. Transient transformation of the sweet osmanthus petals and stable transformation of tobacco (*Nicotiana tabacum*) were used to explore the functions of *OfCYP142*. We anticipate that this research will provide valuable gene resources concerning the CYP family to further reveal the molecular regulation mechanism of the VOCs of *O. fragrans*.

## 2. Results

### 2.1. Identification of OfCYPs in O. fragrans

There were 333 putative *OfCYP*s identified from the genome of sweet osmanthus using HMMER software. Among them, 276 genes were recognized as members of the *P450* gene family by virtue of their heme-binding loop which contains the complete conserved domains (Phe(F)-X-X-Gly(G)-X-Arg(R)-X-Cys(C)-X-Gly(G)). These genes were named *OfCYP1* to *OfCYP276* according to their position on chromosomes. The length of the OfCYP proteins varied from 355 (OfCYP50) to 1183 (OfCYP21) amino acids and the corresponding molecular weights were 40.12 and 135.65 kDa. The theoretical isoelectric point values ranged from 5.32 (OfCYP145) to 9.61 (OfCYP232) (Appendix A).

### 2.2. Phylogenetic Analysis of OfCYPs in O. fragrans

The evolutionary relationships of sweet osmanthus cytochrome P450 proteins were investigated using 276 OfCYP and 264 AtCYP proteins (Appendix A). In light of the classification in Arabidopsis, all OfCYP proteins were divided into 10 clans with two groups, A-type (CYP71 clan) and non-A type (CYP72/711/727/97/86/710/74/85/51 clan) including 141 and 135 OfCYPs, respectively [7,33]. Typical structures of the P450 family, such as heme-binding domain, I-helix region, K-helix domain, and PERF motif were all found in these 276 OfCYP proteins (Figure 1). The largest clan was the CYP71 clan which contained 141 OfCYP proteins, and the smallest was the CYP727 clan, with only one OfCYP protein. In addition, the CYP51, CYP74, and CYP710 clans did not contain any OfCYP proteins (Figure 1).

### 2.3. Gene Structure and Conserved Motif Analysis of OfCYPs

To further investigate gene structure, the structural patterns of introns and exons of *OfCYP*s were analyzed. The gene clans were divided into 40 families (Appendix A). All the *OfCYP*s contained P450 conserved domains named CYPX or P450. Among the 276 *OfCYP*s, the gene containing the most exons had 22 exons, and the least had only one. Exon analysis showed that most members of the same family showed similar exon–intron structural patterns (Appendix A). For example, 25 genes had no introns, 23 of which were concentrated in the CYP86 clan including the CYP86, CYP96, and CYP94 families. The members of the CYP77 and CYP89 families had at most one exon whereas among the CYP97 family members containing exons, the numbers ranged within 10–17. All CYP81 family members had two exons except for *OfCYP212* and *OfCYP221* which had three exons each.

Evaluation of conserved motifs played a vital role in the functional prediction of OfCYP proteins. A sum of 20 conserved motifs (motifs 1–20) were identified in 276 OfCYP proteins using MEME online software (Appendix A). Most members within the same family exhibited similar motif organizations that verified the grouping results. Motif 1 was widely present in all OfCYP proteins except OfCYP157; and all of the OfCYP proteins contained motif 3 except OfCYP52 and OfCYP77. However, motifs 4, 7, and 16 were only present in the CYP71 clan. Motif 19 was only found in the CYP85 clan. In general, motif composition of OfCYP proteins in the same subfamily was similar but the location of motifs of each subfamily member showed marked differences. This similarity of the motif components of the OfCYP proteins within the same family demonstrates that the OfCYP proteins from the same family may perform similar functions, and that some conserved motifs may play important roles in family-specific functions.

### 2.4. Chromosomal Distribution, Duplication Events, and Collinear Analysis of OfCYPs

According to the genome annotation, all 276 *OfCYP*s were mapped to 23 chromosomes, with 2–35 on each chromosome. The most *OfCYP*s were located on Chr1, whereas Chr16 contained only two genes (Figure 2). We identified 67 tandem duplicated genes and 38 of these genes were divided into 19 groups: *OfCYP16/17*, *OfCYP31/32*, *OfCYP38/39*, *OfCYP44/45*, *OfCYP65/66*, *OfCYP75/76*, *OfCYP110/111*, *OfCYP114/115*, *OfCYP131/132*, *OfCYP133/134*, *OfCYP148/149*, *OfCYP151/152*, *OfCYP154/155*, *OfCYP165/166*, *OfCYP192/193*, *OfCYP199/200*, *OfCYP212/213*, *OfCYP234/235*, and *OfCYP263/264*. Twenty-one of them were divided into seven groups: *OfCYP22/23/24*, *OfCYP72/73/74*, *OfCYP86/87/88*, *OfCYP89/90/91*, *OfCYP97/98/99*, *OfCYP116/117/118*, and *OfCYP243/244/245*. In addition, eight were divided into two groups: *OfCYP167/168/169/170* and *OfCYP238/239/240/241*. There were nine tandem duplicated genes on Chr19, which was the chromosome with the most tandem duplicated genes. Moreover, 66 segmental duplication pairs with 163 *OfCYP*s were identified (Figure 3).

### 2.5. Expression Patterns of OfCYPs in Different Tissues

Studies in some plants that have reported the function of *CYP* genes demonstrated that *CYP* genes play a critical role in synthesis and volatilization of VOCs [17,34,35,36]. For more insight into the any role of *OfCYP*s in synthesis of floral VOCs of sweet osmanthus, the expression levels of 276 *OfCYP*s in seven distinct organ samples (root, stem, two distinct stages of leaf development, and three different stages of flower development) were quantified as RPKM (reads per kilobase per million mapped reads) values, obtained from the RNA-seq data of seven distinct tissues of sweet osmanthus [32]. The expression signals of 92.8% (256/276) of sweet osmanthus *CYP* genes were determined. Interestingly, *OfCYP206/227/230/45* showed consistently high expression in all tissues, whereas 34 *OfCYP*s displayed low expression levels (FPKM < 1) in all tissues (Appendix A). *OfCYP220* was expressed only in the flower development stages and its expression significantly increased during stages S2–S5 (bud-eye–flower fading stages) while *OfCYP115/118/180* were expressed only in the root (Figure 4A). The different expression patterns of each *OfCYP* in various tissues potentially indicate the functional divergence of *OfCYP*s in sweet osmanthus.

### 2.6. Expression Patterns Analysis of OfCYPs during Sweet Osmanthus Flower Development

To further expound possible functions of *OfCYP*s in sweet osmanthus, the expression patterns of *OfCYP*s participating in sweet osmanthus flower growth and development were studied using RNA-seq data in five distinct stages of flower development (Appendix A). Among the 276 *OfCYP*s, 154 genes were used to draw a heatmap due to their constitutive expression (FPKM > 1 in at least one sample). *OfCYP206/227/230* still showed consistently high expression in all stages of flower development, whereas *OfCYP45* differed from before and had relatively low expression levels. Interestingly, *OfCYP220* expression initially increased and then decreased, and increased again during stages S3–S4 (primary–full blooming stages) and eventually decreased across the last two stages. The expression levels of *OfCYP240/116/218/156* showed a consistent increasing trend across the five flower development stages, whereas *OfCYP231/210* showed a decreasing trend (Figure 4B).

A total of seven *OfCYP*s, preferentially expressed in flowers, were selected as candidate genes, referring to the expression pattern of the *OfCYP*s derived from the RNA-seq. The differential expressions of the candidate genes were verified at five flower stages in sweet osmanthus using qRT-PCR (Figure 5A,B). The expression levels of various candidate genes at different stages exhibited significant differences. *OfCYP13/227* exhibited similar stage-specific expression patterns, specifically expressed in S2 and S4, but had relatively low expression in other stages. *OfCYP142/220/39* showed significantly higher expression during S4–S5 than in other stages. As a whole, relative expression trends of *OfCYP*s were consistent with the expression trends in the RNA-seq data in five distinct stages of flower development. Correlation analysis revealed that the expression pattern of *OfCYP142* showed the highest positive correlation with the expression pattern of *OfTPS2* (Figure 5C).

### 2.7. Expression Patterns of OfCYPs under Abiotic Stress

To further explore the role of *OfCYP*s in response to cold and salt stresses, RNA-seq data under abiotic stress were employed to investigate the expression levels of *OfCYP*s in *O. fragrans* under cold (4 °C) or salt treatment (Appendix A). Among the 276 *OfCYP*s, 170 and 175 genes showed expression (FPKM > 1 in at least one sample) under cold and salt treatments, respectively. After cold treatment (4 °C) for 5 d, some *OfCYP*s were upregulated, but the highest expression level was either at 12 h (*OfCYP142*), 24 h (*OfCYP196*), 72 h (*OfCYP248*), or 120 h (*OfCYP262*) in response to cold stress. *OfCYP2/131/132/134/171/171* and some others were obviously suppressed by cold stress and their expression levels showed an increasing trend after recovery for 72 h (Figure 6A). Compared to controls, expression of *OfCYP23/39/49/272/138* and some other genes was upregulated after salt treatment for 72 h; however, expression of *OfCYP195/237* and some other genes was clearly downregulated (Figure 6B). These results suggested that some *OfCYP*s might be involved in cold and salt stress responses in *O. fragrans.* Remarkably, *OfCYP142* was upregulated not only after cold treatment (4 °C) for 12 h but also after salt treatment for 6 and 72 h.

### 2.8. Transient Expression of OfCYP142 in Flowers Altered Contents of VOCs

To further explain the potential roles of the candidate *OfCYP*s in regulating the synthesis of the volatile metabolic components, *OfCYP142* was transiently expressed in the flowers of sweet osmanthus (Figure 7B). The qRT-PCR results demonstrated that the relative expression level of *OfCYP142* in transiently transformed flowers was well above that in the control (empty vector) (Figure 7A). The PCA-X and OPLS-DA plots (Figure 7C,D) showed obvious metabolic differences in the flowers overexpressing *OfCYP142* compared with those transformed with the empty vector. Generally, monoterpene volatiles, especially linalool and linalool derivatives, are important contents of volatile aroma compounds in *O. fragrans*. Interestingly, compared with the aroma contents in flowers transformed with the empty vector, the linalool contents in flowers overexpressing *OfCYP142* showed significant differences according to the GC–MS analysis with VIP > 1 and *p* < 0.05 tested by SPSS software (Figure 7E–G and Appendix A).

### 2.9. Identifying Differentially Expressed Genes (DEGs) in Plants with Transient Expression of OfCYP142

Transcriptome profiling of the transgenic flowers and flowers transformed with the empty vector was performed to facilitate a deeper understanding of the possible molecular mechanisms of *OfCYP142* in stimulating linalool synthesis (Appendix A). Use of criteria of expression change >2-fold and *p* < 0.05 showed that 1079 genes were DEGs in the transgenic flowers compared with flowers transformed with the empty vector (Figure 8A). The DEGs in the transgenic flowers were clustered into 19 categories and enriched in 94 pathways in accordance with the Kyoto Encyclopedia of Genes and Genomes (KEGG) (Appendix A). Some overrepresented categories included genes involved in metabolism of terpenoids and polyketides, signal transduction, membrane transport, biosynthesis of other secondary metabolites, and environmental adaptation, implying that *OfCYP142* overexpression may affect synthesis of monoterpene compounds and stress signaling and response (Figure 8C). Among the DEGs, expression of *OfSDR1* was markedly upregulated. Additionally, expression of some other genes in the 2-C-methyl-D-erythritol 4-phosphate (MEP) pathway including *OfDXS2*, *OfDXR*, and *OfTPS2* was upregulated while *OfGGPS1* was downregulated in transgenic flowers. Their expression patterns were further validated by qRT-PCR (Figure 8B).

### 2.10. Overexpression of OfCYP142 Promoted VOC Accumulation in Transgenic Tobacco Petals

For further confirm the function of *OfCYP142*, we ectopically expressed *OfCYP142* in multiple transgenic lines of tobacco. Positive transgenic lines were screened by RT-PCR, indicating that *OfCYP142* had been integrated into the tobacco genome and expressed successfully (Figure 9A). Lines 5, 10, and 11 were selected for the following analysis because of their consistent growth status and flowering period. The transgenic plants showed no obvious changes in appearance, such as flower shape and color, compared with tobacco transformed with the empty vector, but the aroma contents in flowers significantly differed (Figure 9B–D and Appendix A). As expected, the linalool content was remarkably higher in transgenic than control flowers (Figure 9E). In addition, compared to controls, the contents of some important VOCs in tobacco such as leaf alcohol (with fragrant smell) and sesquiterpene (-)-caryophyllene oxide also increased significantly in transgenic plants (Figure 9F–H).

## 3. Discussion

The *CYP* gene family has been reported in numerous studies of some plant species with genome-wide characterization [7,37]. The number of *CYP* genes ranges from 79 in cotton to 355 in durian (Appendix A). In the present study, 276 *OfCYP*s were identified from the *O. fragrans* reference genome and clustered into seven clans, which were further categorized into 40 families (Appendix A). There were fewer *CYP* genes in *O. fragrans* than in most woody plants. Gene-duplication events, especially tandem duplication, are considered to be one of the main reasons for gene family expansion during the process of evolution [38]. A total of 67 tandem duplicated genes and 66 segmental duplication pairs with 163 *OfCYP*s were identified in our study (Figure 3), indicating that evolution of the *OfCYP* family may also have been driven by gene-duplication events.

The conserved heme-binding domain (FxxGxRxCxG) and the PERF consensus (PXRX) are P450 signature motifs [39]. Motifs 1 and 3, representing the heme-binding domain and the PERF consensus, respectively, were present in every OfCYP protein. Most OfCYP proteins within the same clan exhibited similar motif components but with significant differences among clans (Appendix A), indicating that the OfCYP members within the same family may have similar functions, and that some motifs may play a vital role in family-specific functions. The similar motif distribution patterns within the same family also imply that the identification and division of *OfCYP* gene families in this study is reliable.

Previous studies have shown that *CYP* genes are widely distributed and participate in the synthesis of many metabolites, especially secondary metabolites [40]. It is well known that genes expressed specifically in certain tissues can have tissue-specific functions. The analysis of RPKM values from the transcriptome data from seven different organs provided organ-specific expression data for the *OfCYP*s (Appendix A), implying that these genes may play different roles in various organs in *O. fragrans*. *AtCYP73A5* which plays an essential role in phenylpropanoid metabolism, was expressed more in flowers than other organs [41]. In our study, *OfCYP227*, which has high sequence identity to *AtCYP73A5,* showed consistently high expression in all organs and was extremely highly expressed in flowers. Seven genes similar to *OfCYP227*, which were specifically expressed in blooming flowers, were selected to verify their expression patterns (Figure 5A). The results indicated that these *OfCYP*s had high expression levels in flowers and may play a critical role in the synthesis and emission of fragrance by *O. fragrans* flowers.

The expression of genes involved in stress responses usually responds differently to different stresses [42]. The analysis of RNA-seq data under abiotic stress showed that some *OfCYP*s were upregulated, but the highest expression level was either at 12 h (*OfCYP142*), 24 h (*OfCYP196*), 72 h (*OfCYP248*), or 120 h (*OfCYP262*) in response to cold stress. *OfCYP142* was also upregulated after salt treatment for 6 and 72 h (Figure 6)–thus this gene was induced by both cold and salt stress. In addition, *OsCYP94C2b* was demonstrated to endow rice with salinity tolerance [21]. *OfCYP142* has very high homology with *OsCYP94C2b*, further confirming that *OfCYP142* was induced by salt stress.

Previous studies have shown that CYP proteins that participate in monoterpenoid metabolism are usually co-expressed with terpene synthases [16,43], and gene co-expression analysis has been used to predict which *P450* genes are involved in monoterpenoid metabolism [16]. *OfTPS2* was reported to play an important role in the synthesis of linalool in *O*. *fragrans* [44]. To gain insight into the functions of *P450* genes in linalool synthesis and metabolism, the relationships between the expression patterns of *OfCYP*s and *OfTPS2* were investigated (Figure 5C). *OfCYP142* which had an expression pattern with the highest positive correlation with *OfTPS2*, was selected as a candidate gene to further explore its function. Consistent with the above predictions, *OfCYP142* overexpression in *O*. *fragrans* petals promoted linalool accumulation (Figure 7C), implying that *OfCYP142* may play a critical role in stimulating the synthesis of linalool. Linalool is formed by a dephosphorylation reaction and a hydroxylation reaction of geranyl pyrophosphate (GPP) catalyzed by TPS [45]. Many *CYP* genes have been reported to be involved in the process of linalool hydroxylation conversion to linalool oxides [16,17]. We speculate that *OfCYP142* may be involved in the hydroxylation reaction of GPP to promote the synthesis of linalool.

A previous study demonstrated that expression of several MEP monoterpene substrate pathway genes preceding terpene formation was positively correlated with corresponding downstream volatile terpenes [46]. Transcriptome profiling of the transgenic flowers indicated that many key regulation genes in the MEP pathway showed an upregulated expression trend (Figure 8B). The co-ordinated regulation relationships are found in most of MEP pathway genes [47]. We speculate the overexpression of *OfCYP142* could trigger some co-ordinated regulation mechanisms, and upregulate the expression levels of corresponding MEP pathway genes which can promote the floral VOCs accumulation. The results of tobacco transformation further verified the above conclusion, with *OfCYP142* overexpression significantly elevating some floral VOCs including linalool, leaf alcohol, and (-)-caryophyllene oxide in tobacco flowers (Figure 9E–H). Leaf alcohol is present in a wide range of foods, beverages, and fine fragrances, due to its characteristic intense smell of fresh cut grass or leaves [48,49]; and (-)-caryophyllene oxide is a major representative sesquiterpenoid, and an important resource for organic and medicinal chemistry [50,51]. Many *CYP* genes have been reported to participate in sesquiterpene biosynthesis [52]. These results imply that *OfCYP142* is not only involved in linalool biosynthesis but can also promote accumulation of the other floral VOCs.

## 4. Materials and Methods

### 4.1. Plant Materials

The *O. fragrans* plants used were cultivated in the fields of Nanjing Forestry University, Nanjing, China. Petals of ‘RiXianggui’ at five stages (S1, bud-pedicel stage; S2, bud-eye stage; S3, primary blooming stage; S4, full blooming stage; and S5, flower fading stage) were collected during 9:00–11:00, frozen in liquid nitrogen, and stored in a freezer at −80 °C. Fresh petals of ‘Sijigui’ at full blooming stage were collected, then washed with pure water, and finally used for transient transformation. A total of 0.3 g of petals were randomly picked from two *O. fragrans* trees as one replicate, and three biological replicates were collected per sampling.

### 4.2. Bioinformatics Analysis of the P450 Family in O. fragrans

The hidden Markov model profile of the P450 domain (PF00067) was downloaded from Pfam online (https://pfam.xfam.org/) (accessed on 15 August 2020) and used to retrieve the protein sequence of sweet osmanthus with HMMER (v3.0). The candidate *P450* genes were further uploaded to the CDD (http://www.ncbi.nlm.nih.gov/Structure/cdd/wrpsb.cgi) (accessed on 15 August 2020) online website to confirm whether they contained the appropriate P450 domain. A sum of 264 Arabidopsis P450 proteins and 276 putative OfCYP proteins were aligned using MUSCLE, and a phylogenetic tree was constructed in MEGA 7.0 by the NJ approach, with 1000 bootstrap replicates. The P450 protein sequences in Arabidopsis were downloaded from TAIR (https://www.arabidopsis.org/) (accessed on 12 September 2020). The online WoLF PSORT site (https://wolfpsort.hgc.jp/) (accessed on 24 August 2021) was employed to forecast the subcellular localizations of OfCYP proteins. The Sequence Structure Illustrator program in TBtools was employed to visualize the gene structures and coding sequences of *OfCYP*s [53]. The MEME Suite webserver was employed to predict conserved motifs of *OfCYP* family members identifying a maximum of 20 motifs [54]. All *OfCYP*s were mapped to sweet osmanthus chromosomes based on the sweet osmanthus genomic GFF3 file. Multiple collinear scanning toolkits (MCScanX) with the default parameters were employed for analysis of *OfCYP* gene-duplication events [55]. The transcription levels of *OfCYPs* in leaves of two-year-old cutting-based seedlings of *O. fragrans* (‘Rixianggui’) under cold and salt stress conditions were obtained from our previous transcriptome data (unpublished). The expression profiles of *OfCYPs* in response to cold and salt stresses were visualized in a heatmap constructed using TBtools.

### 4.3. RNA Isolation and Quantitative Real-Time PCR (qRT-PCR) Analysis

The buds and flowers of sweet osmanthus were collected at five developmental stages: S1–S5. Isolation of total RNA was carried out by the RNAprep Pure Plant Kit (Tiangen, Beijing, China). Complementary DNA (cDNA) was synthesized by One-Step gDNA Removal and cDNA Synthesis Super Mix (Transgen, Beijing, China). Primer Premier 5.0 was employed to design the qRT-PCR primers of the *OfCYP*s. Genes *OfRAN* and *NtEF-1α* (accession no: AF120093) were used as normalizers (Appendix A). Each qRT-PCR assay consisted of three biological samples and each biological sample had three technical replicates.

### 4.4. Transient Overexpression of OfCYP142 in O. fragrans Petals

To explore the function of *OfCYP142* in the synthesis and metabolization of volatile components, the overexpression vector of *OfCYP142* was used for transient transformation of sweet osmanthus petals using a published method with some modifications [56]. The *OfCYP142* coding region was amplified with specific primers (Appendix A) and inserted into the pSuper1300-GFP vector. Then the vector was transformed into *Agrobacterium tumefaciens* strain GV3101.

After the overnight culture of Agrobacterium, the cultured products were poured into tubes and then spun down at 2716× *g* for 10 min at 4 °C. A solution of 10 mM MgCl_2_, 10 mM MES, and 150 mM acetosyringone was used to resuspend the Agrobacterium cultures until an OD_600_ of 0.6–0.8 was reached. The full blooming petals of ‘Sijigui’ were wrapped with gauze and then vacuum infiltrated with Agrobacterium at −0.075 MPa for 15 min. They were returned to normal air pressure and then 15 min of vacuum infiltration was repeated. After infiltration, the petals were removed and placed on 0.5% agar medium in darkness for 48 h.

### 4.5. Transformation of OfCYP142 in Tobacco

The *O. fragrans* stable transformation was performed by a published method with some modifications [57]. The young leaves of tobacco cv. K326 plants were first washed with pure water and then sterilized on an ultra-clean workbench. The *OfCYP142* coding region was amplified with specific primers (Appendix A) and inserted into the PBI121-GUS vector. The vector was transformed into the *Agrobacterium tumefaciens* strain GV3101. Agrobacterium cultures were the same as those used for transient transformation of *O. fragrans*. When culture OD_600_ of 0.5 was reached, it was used for transformation. The explants were first soaked in 75% ethanol for 30 s and then immediately washed three times with sterile pure water. After washing, explants were placed in 5% NaClO for 6 min and then rinsed with sterile pure water five to six times. A sterile scalpel was used to excise the leaf margins and veins of the explants, and the rest of the leaves were cut into disks with a diameter of 1 cm and infiltrated with Agrobacterium for 10 min.

After 48 h of co-cultivation in darkness on the symbiotic medium, the leaf disks were transferred into selection medium containing 2.0 mg/L 6-benzylaminopurine, 0.1 mg/L 1-naphthaleneacetic acid, 100 mg/L kanamycin, and 400 mg/L cefotaxime. The medium was kept under conditions of 25 °C with 16 h of light per day, and changed every 15 days to ensure nutrient supply. When the shoots regenerated and the mature leaves emerged, the independent kanamycin-resistant tobacco plants were then selected and transferred to a rooting medium. When the fibrous roots of plantlets were developed and mature, plantlets were removed from the medium and planted in flowerpots, and grown in a greenhouse at Nanjing Forestry University. The control plants were transformed with an empty vector and also regenerated under the same conditions.

### 4.6. Aroma Compound Analysis of Transgenic Plants

The headspace SPME (solid-phase microextraction) method was used to collect the volatile components. Each 0.3-g sample of transgenic *O. fragrans* petals to be tested was added as a biological replicate to a 4-mL extraction vial, and all transgenic lines were collected in three biological replicates. Ethyl caprate was diluted 3000-fold with methanol, and 1 μL of the diluted product was added to the side of the vial at 3 cm from the bottom (Macklin Inc., Shanghai, China). The SPME fiber of DB-5 MS was inserted into the capped vial containing the sample, then the vial was heated in a constant temperature water bath at 45 °C for 35 min for extraction. After extraction, the SPME fiber was injected into the heated syringe port for 3 min. Desorption was carried out at 250 °C. The carrier gas was high-purity helium with a flow rate of 1 mL/min and a split ratio of 10:1 (Supelco Inc., Bellefonte, PA, USA).

Later, the desorbed aroma compounds were identified using Trace DSQ gas chromatography–mass spectrometry (GC–MS) equipment, and the RIL values of volatile compounds were calculated using N-alkanes, which were compared with the RIL values in DB-5MS to verify the identification of aroma compounds detection [58]. Differences in relative content of aroma metabolites of petals with transient expression were detected using orthogonal partial least squares discriminant analysis (OPLS-DA) in SIMCA 14.1 software (Umetrics AB, Umea, Sweden). Each 30-mL SPME vial with two transgenic tobacco flowers was used to measure the volatile content of plants that were stably transformed, and the calculation of the relative volatile content of transgenic tobacco flowers was normalized according to the sample weight. The capped vials were placed in water at 60 °C for 30 min during extraction. Other conditions were the same as the those for the transient transformation of *O. fragrans*.

## 5. Conclusions

In summary, 276 *OfCYP*s were identified in the *O. fragrans* genome. They were divided into 40 families, according to their evolutionary characteristics and genetic structure. Tandem duplication and segmental duplication events were shown to play essential roles in the expansion of the *OfCYP* gene family. The RNA-seq data suggested that many *OfCYP*s were tissue-specifically expressed, and some were induced by abiotic stresses. Here, a total seven of *OfCYP*s that were preferentially expressed in blooming flowers were identified. Among them, *OfCYP142* could effectively promote the accumulation of the floral VOCs of plants through influencing the MEP pathway. This study provides comprehensive information on *OfCYP*s and useful gene resources for the molecular breeding of ornamental plants. Follow-up research on these findings may be used to improve flavor quality of *O. fragrans* petals.

## Figures and Tables

**Figure 1 ijms-23-12150-f001:**
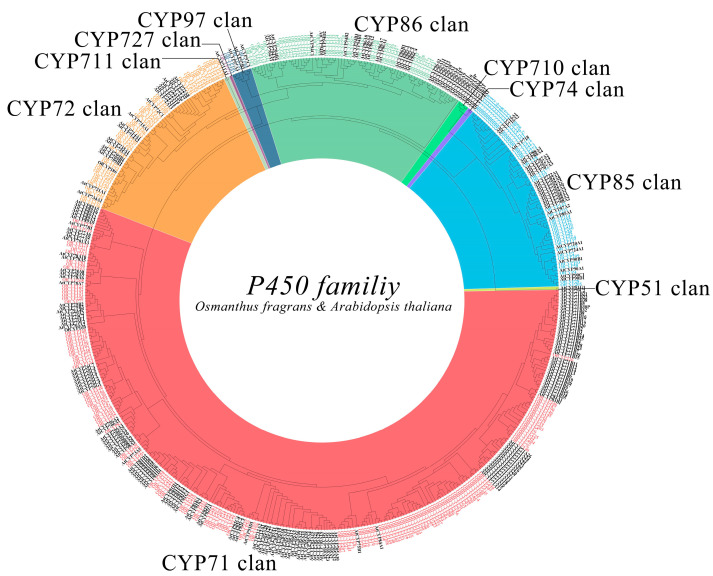
Phylogenetic analysis of 276 OfCYP and 264 Arabidopsis (black labels) AtCYP proteins. The ClustalW program was used to perform alignment, and then all proteins were imported into MEGA 7.0 to build the phylogenetic tree by the neighbor-joining approach with 1000 bootstrap iterations.

**Figure 2 ijms-23-12150-f002:**
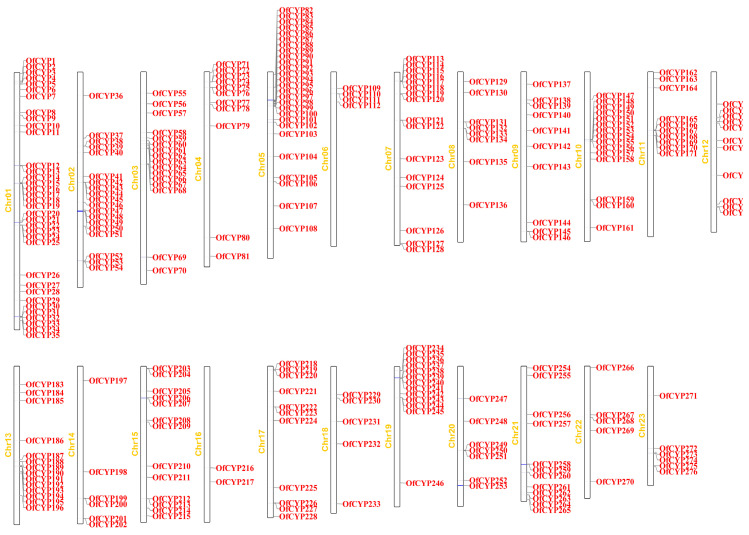
Chromosomal distribution of *P450* genes in *O. fragrans*. The left scale indicates the length (Mb, megabases) of *O. fragrans* chromosomes.

**Figure 3 ijms-23-12150-f003:**
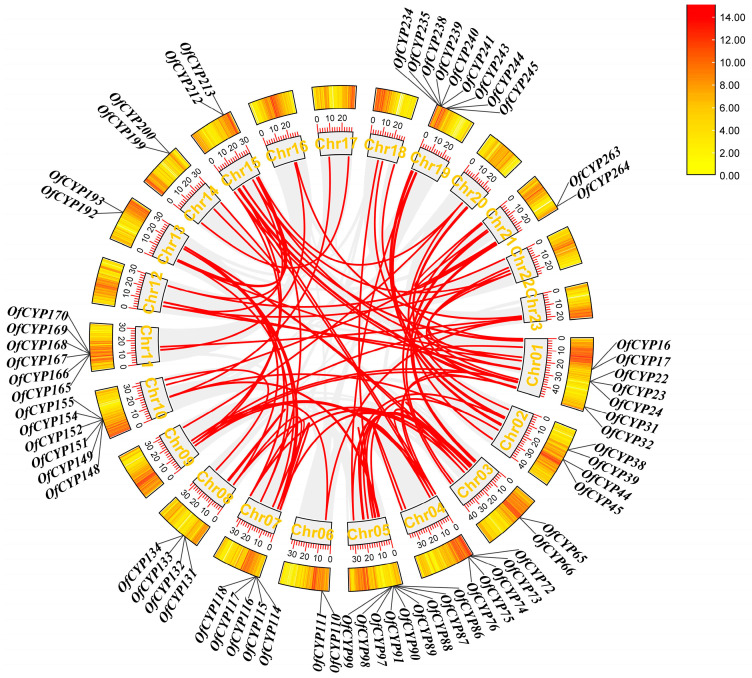
The gray lines in the background represent all syntenic blocks while the marked red lines represent 66 segmental duplication pairs of *OfCYP*s in the sweet osmanthus genome. The genes marked in the outermost layer of the circle are 67 tandem duplicated genes. The numbers on the tick marks above the chromosome names indicate the number of *CYP* genes contained on each chromosome. The density of genes on each chromosome is represented by the heatmap in the orange rectangle.

**Figure 4 ijms-23-12150-f004:**
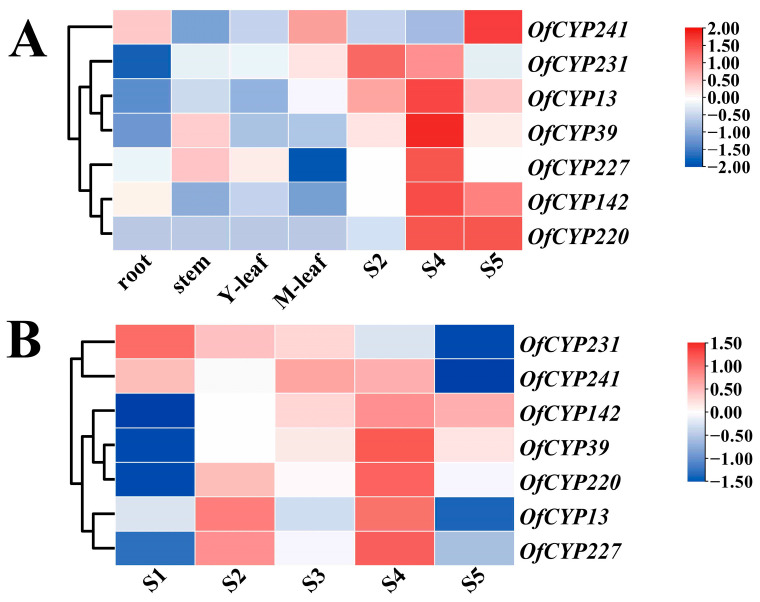
Expression profiles of key *OfCYP*s screened in different tissues and at different stages of flower development. (**A**) Transcription levels of key *OfCYP*s screened in different tissues. The different tissues were root, stem, and leaf (Y-leaf, young leaf; M-leaf, mature leaf); flower (S2, bud-eye stage; S4, full blooming stage; S5, flower fading stage). (**B**) Transcription levels of key *OfCYP*s screened in five distinct stages of flower development: S1, bud-pedicel stage; S2, bud-eye stage; S3, primary blooming stage; S4, full blooming stage; and S5, flower fading stage. The RPKM values obtained from the transcriptome data and FPKM (fragments per kilobase of exon model per million mapped fragments) values derived from the RNA-seq data in five distinct stages of flower development of *OfCYP*s were normalized by log_2_ transformation. The heatmap was generated by TBtools software.

**Figure 5 ijms-23-12150-f005:**
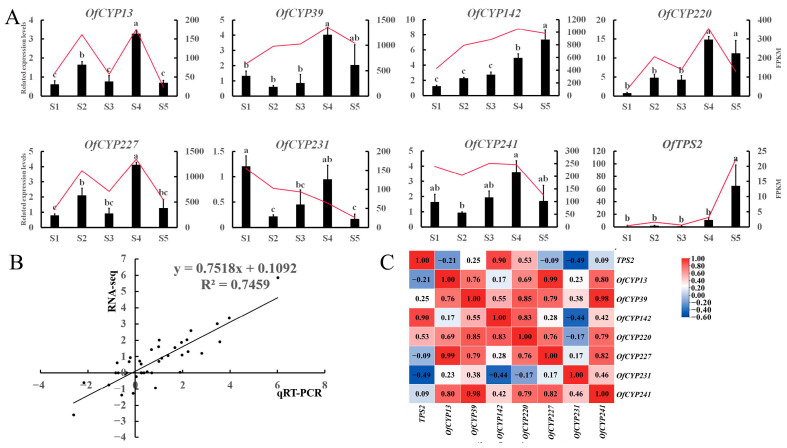
(**A**) Transcription levels of *OfCYP*s in different stages of sweet osmanthus flowers: S1, bud-pedicel stage; S2, bud-eye stage; S3, primary blooming stage; S4, full blooming stage; and S5, flower fading stage. The levels of transcription are the mean values of three biological samples, each sample had three technical replicates. The standard error (SE) was used to draw error bars. The different letters marked above the error bars denote the results of the significant differences analysis at *p* < 0.05 obtained using SPSS (version 25). (**B**) Correlation analysis chart of qRT-PCR and transcriptome data FPKM values. (**C**) Correlation analysis of expression of *OfCYP*s and *OfTPS2*. Correlation coefficients from negative to positive correlation are indicated by a gradient from blue to red.

**Figure 6 ijms-23-12150-f006:**
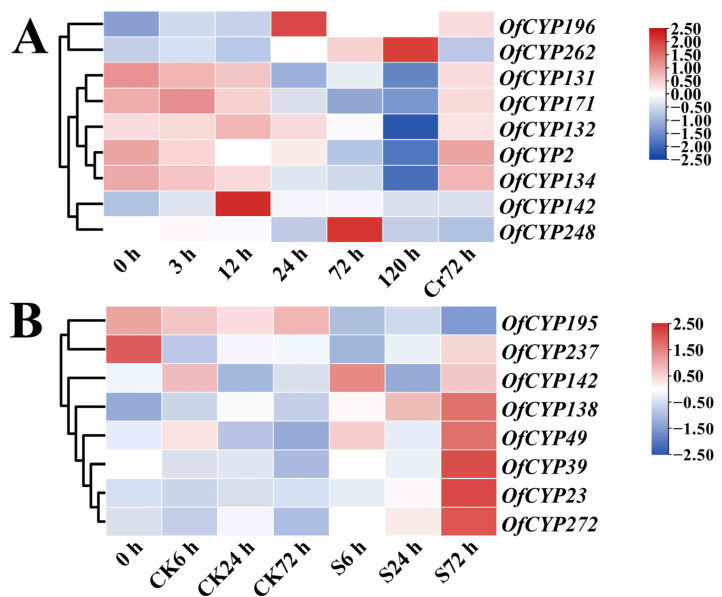
Expression levels of key *OfCYP*s screened in response to cold and salt stress conditions. (**A**) Expression profiles of *OfCYP*s in response to cold stress. 0 h, 3 h, 12 h, 24 h, 72 h, and 120 h represent the different times that the cold treatment was maintained. The Cr72 h represents recovery for 72 h after a 120-h cold treatment. (**B**) Expression profiles of *OfCYP*s in response to salt stress. S0, S6, S24, and S72 denote salt stress for up to 0 (control), 6, 24, and 72 h, respectively. The FPKM values for *OfCYP*s in leaves were obtained from the sweet osmanthus RNA-seq data under cold and salt stress conditions (Appendix A) and were normalized by log_2_ transformation. The heatmap was generated by TBtools software.

**Figure 7 ijms-23-12150-f007:**
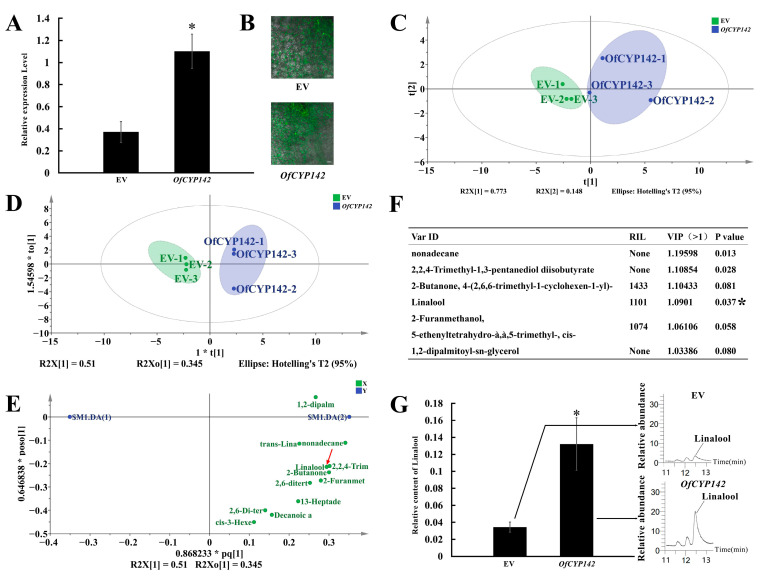
Functional characterization of *OfCYP142* by transient expression. (**A**) qRT-PCR analysis of the *OfCYP142* relative expression levels in control and transgenic *O. fragrans* flowers. (**B**) The GFP fluorescence of p35S:OfCYP142:GFP and empty vector fusion proteins. (**C**) Score scatter plot of the flower GC–MS profiles of two types of petals with transient expression of *OfCYP142* and empty vector in *O. fragrans*, determined with PCA-X analysis in SIMCA14.1. Each type of petals consisted of three replicates. (**D**) Score scatter plot of the flower GC–MS profiles of two types of petals with transient expression of *OfCYP142* and empty vector in *O. fragrans*, determined with OPLS-DA analysis in SIMCA14.1. (**E**) Loading scatter plot analysis of floral volatile organic compounds in petals after transient transformation with 35S:*OfCYP142*:GFP and empty vectors. The red arrow marks the linalool. (**F**) Information of key aromatic components selected based on VIP from OPLS-DA (VIP > 1). (**G**) Relative content analysis of linalool after transient transformation with 35S:*OfCYP142*:GFP and empty vectors. “*” represents significant difference at *p* < 0.05.

**Figure 8 ijms-23-12150-f008:**
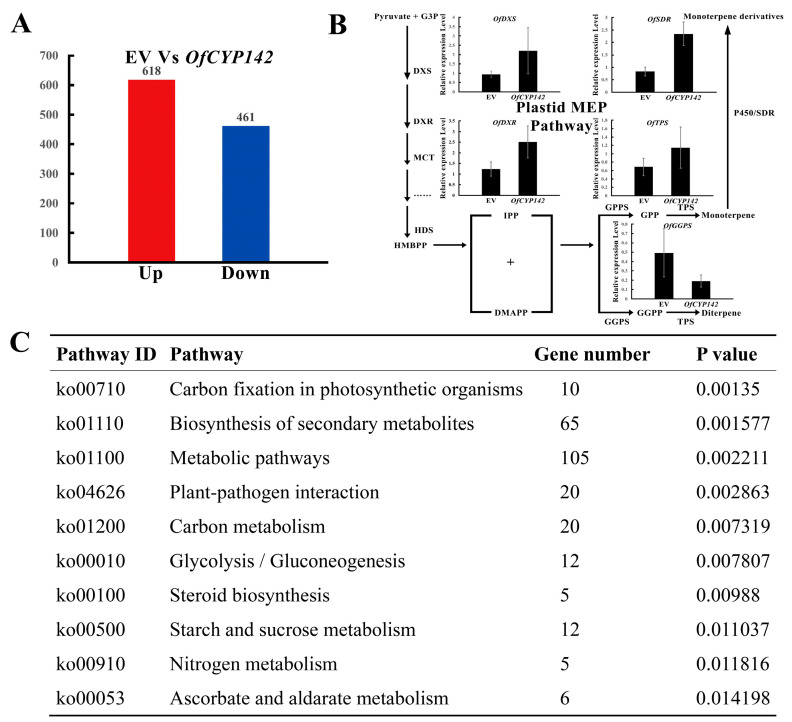
Transcriptome profiling of transgenic flowers. (**A**) Number of DEGs in transgenic flowers. (**B**) The MEP pathway responsible for the formation of terpenes in plants, and qRT-PCR validation of expression patterns of candidate genes in the MEP pathway. Relative expression levels of the selected genes were analyzed by qRT-RCR using *OfRAN* as an internal control. (**C**) Information of top 10 enriched KEGG terms according to *p*-value.

**Figure 9 ijms-23-12150-f009:**
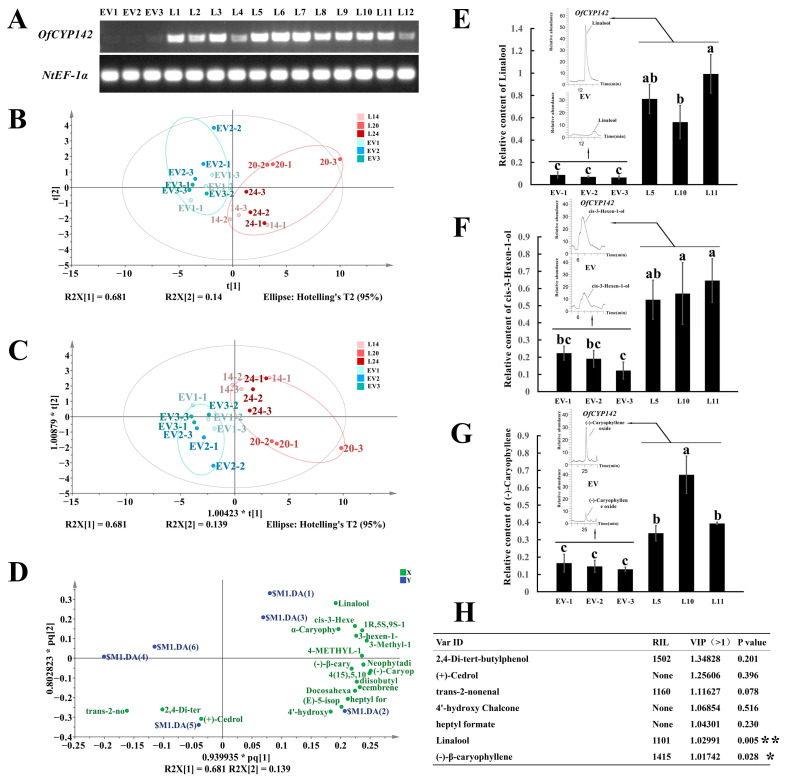
Functional characterization of *OfCYP142* by stable transformation. (**A**) Semiquantitative RT-PCR to determine the expression intensity of *OfCYP142*; a fragment of the *NtEF-1α* gene was amplified as the normalizer, EV1–3 denote different plant lines that were stably transformed with empty vectors, and L1–12 denote different plant lines that were stably transformed with 35S:*OfCYP142*:GUS vectors. (**B**) Score scatter plot of the flower GC–MS profiles of two types of flowers with overexpression of *OfCYP142* and empty vector in tobacco, determined with PCA-X analysis in SIMCA14.1. Each type of plants consisted of three replicates. (**C**) Score scatter plot of the flower GC–MS profiles of two types of flowers with overexpression of *OfCYP142* and empty vector in tobacco, determined with OPLS-DA analysis in SIMCA14.1. (**D**) Loading scatter plot analysis of floral volatile organic compounds in plants that were stably transformed with 35S:*OfCYP142*:GUS and empty vectors. (**E**) Floral composition analysis of linalool after stable transformation with 35S:*OfCYP142*:GUS and empty vectors. (**F**) Floral composition analysis of leaf alcohol after stable transformation with 35S:*OfCYP142*:GUS and empty vectors. (**G**) Floral composition analysis of (-)-caryophyllene oxide after stable transformation with 35S:*OfCYP142*:GUS and empty vectors. (**H**) Information of key aromatic components selected based on VIP from OPLS-DA (VIP > 1); * significant difference at *p* < 0.05 and ** significant difference at *p* < 0.01. The lowercase letters above the columns represent the significant differences as assessed by Duncan’s test at the 0.05 level.

## Data Availability

All data in this study could be found in the manuscript or Appendix A.

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
