# Peer review of "Insights into the Cytochrome P450 Monooxygenase Superfamily in Osmanthus fragrans and the Role of OfCYP142 in Linalool Synthesis"

_ijms, 2022, doi:10.3390/ijms232012150_

Round 1

Reviewer 1 Report

This work aims at identifying the P450 superfamily genes in Osmanthus fragrans and uncovering the functions of these genes. The authors performed a good work. In this manuscript, a total of 276 P450 genes were identified belonging to 40 families. The gene expression profiles of these OfCYPs in different tissues and treatments were analyzed by RNA-seq. Several key genes ( OfCYP142 et al.) were identified. And the function of OfCYP142 was verified in tobacco and O. fragrans. This study provided valuable information about the functions of P450 genes. Here, I give some suggestions for this MS.   

1. In “Results” section: adjust the order of 2.6 section and 2.7 section, so that the logic will be better.  

2. Figure 4 and 5 are too larger to see the detail. I suggested move these two figures to supplementary materials. At the same time, the expression profiles or heatmap of key genes screened can be added in the manuscript to highlight the expression module of key or important genes, so as to facilitate readers to obtain more effective information.

Reviewer 2 Report

The manuscript "Insights into the cytochrome P450 monooxygenase superfamily in Osmanthus fragrans and the role of OfCYP142 in linalool synthesis" investigated the P450 superfamily in Osmanthus fragrans at a genome-wide level by checking their phylogeny relationship, chromosomal distribution and RNA expression pattern. Interestingly, the authors also found that CYP94C subfamily member OfCYP142 had the highest positive correlation with the linalool synthesis gene OfTPS2. The transient expression of OfCYP142 in O. fragrans petals suggested that OfCYP142 can increase the content of linalool. In general, this is an interesting paper and most of the results make sense to me. My question or concern is that the identification of DEGs in plants with transient expression of OfCYP142. The authors found that the transient expression of OfCYP142 upregulates the MEP pathway genes. But OfCYP142 belongs to the cytochrome P450 monooxygenase superfamily, I’m a little confused about why this gene act as a transcription regulator. I admitted my confusion might be due to my little knowledge in this area.  

Please see below for more questions and suggestions. 

The text in Figures 1, 4, and 5 is too small to see. You might need to increase the font size. You might also need to provide a high resolution of Figure S1. 

In Figure 1, it might be good to use a line to indicate where the CYP51 clan in the phylogenetic tree, 

In Figure 2, the number of “30” on the tick marks on chromosome 11 is upside down.  

Line 283, Plants with Transient Expression of OfCYP142. 

Line 201, what’s the tissue used for the RNA-seq analysis? There is no description in the Method part as well.  

In Figure 7C, D, and E, the label size in the score scatter plots should be increased a bit. Same issue in Figure 9B, C and D.
